# Yoga an effective strategy for self-management of stress-related problems and wellbeing during COVID19 lockdown: A cross-sectional study

Pooja Swami Sahni[1]*, Kamlesh Singh[1,2☯], Nitesh Sharma[1☯], Rahul Garg[1,3,4☯]

**1** National Resource Centre for Value Education in Engineering, Indian Institute of Technology Delhi, Noida, India, **2** Department of Humanities and Social Sciences, Indian Institute of Technology Delhi, New Delhi, India, **3** Department of Computer Science and Engineering, Indian Institute of Technology Delhi, New Delhi, India, **4** Amar Nath and Shahsi Khosla School of Information Technology, Indian Institute of Technology Delhi, New Delhi, India

☯ These authors contributed equally to this work.
* pooja.sahni@nrcvee.iitd.ac.in

**Data Availability Statement:** The data underlying this study are available on OSF (DOI: osf.io/q7xy9).

**Funding:** The author(s) received no specific funding for this work.

## Abstract

This cross-sectional research aims to study the effect of yoga practice on the illness perception, and wellbeing of healthy adults during 4–10 weeks of lockdown due to COVID19 outbreak. A total of 668 adults (64.7% males, M = 28.12 years, SD = 9.09 years) participated in the online survey. The participants were grouped as; yoga practitioners, other spiritual practitioners, and non-practitioners based on their responses to daily practices that they follow. Yoga practitioners were further examined based on the duration of practice as; long-term, mid-term and beginners. Multivariate analysis indicates that yoga practitioners had significantly lower depression, anxiety, & stress (DASS), and higher general wellbeing (SWGB) as well as higher peace of mind (POMS) than the other two groups. The results further revealed that the yoga practitioners significantly differed in the perception of personal control, illness concern and emotional impact of COVID19. However, there was no significant difference found for the measure of resilience (BRS) in this study. Yoga practitioners also significantly differed in the cognitive reappraisal strategy for regulating their emotions than the other two groups. Interestingly, it was found that beginners -those who had started practicing yoga only during the lockdown period reported no significant difference for general wellbeing and peace of mind when compared to the mid- term practitioner. Evidence supports that yoga was found as an effective self- management strategy to cope with stress, anxiety and depression, and maintain wellbeing during COVID19 lockdown.

## Introduction

A report by the World Economic Forum estimates that about 2.6 billion people around the world have been in some kind of lockdown that may lead to second form of stress-related disorder epidemic in the second half of 2020 [1]. Similar to the World economic forum

**Competing interests:** No authors have competing interests.

estimations, a survey by the Indian Psychiatric Society shows that two-fifth of the people are experiencing common mental disorders, due to lockdown and the prevailing COVID19 pandemic in India [2]. This indicates the need for an urgent action to reduce the adverse effects of the COVID19 lockdown on the general well-being of people.

Various factors have been suggested to be contributing to the worsening of mental health. One of the major factors reported causing stress, anxiety and depression is fear of getting infected with the virus/disease (COVID19) [3]. Previous studies examining illness perception in the context of other chronic diseases such as; diabetes, AIDS and myocardial infarction found that people create their representations of the illness related to its risk of contracting, cause, time the illness will last for, and the consequence of the illness [4–6]. Further, it is also suggested that beliefs plays an important role in the way people create notions for the controllability and cure of the illness [7]. These representations and notions are argued to determine the stress response, and the ways of coping, which is believed to affect the wellbeing [8]. On the other hand, some of the factors that influences the perceived effects of the illness are suggested to be the physical and mental health status of the individuals, with healthy people reporting less cognitive and emotional representations of illness [9]. Even the type of treatment and the preventive measures received are suggested to drive the illness coherence/ understanding and perception about personal control over the illness [10].

Even though there are some studies supporting that yoga can be used as complementary and alternate therapy for mental health, there is need for empirical research studies to provide evidence for yoga as effective strategy for self-management of stress-related problems during COVID19. Further, to the best of our knowledge the empirical investigation for the effects of yoga and other spiritual practices on illness perception and wellbeing related problems experienced by people during COVID19 has not been examined so far. The present research uses a cross-sectional study design to examine the effect of the practice of yoga and other spiritual practices on illness perception, and wellbeing of adults. In this study, wellbeing has been assessed through measures of depression, stress, anxiety, resilience, peace of mind and the strategies employed to regulate the emotional upheavals. This approach has been reported in earlier studies that have examined wellbeing in terms of anxiety, stress, and depression [11–13], emotion regulation [14]and as a measure of peace of mind [15]. Wellbeing has also been shown to positively correlate with resilience [16]

Previous research has suggested that yoga can be used as a non-pharmaceutical measure or as a complement to drug therapy for treatment or cure of modern epidemic diseases like mental stress, obesity, diabetes, hypertension, coronary heart disease, and chronic obstructive pulmonary disease [17]. Some recent studies also propose that yoga can assume a groundbreaking complementary and alternative therapy in the battle against the novel coronavirus while improving the physical and mental wellbeing of people in this pandemic circumstance [18, 19]. However, for long researchers have debated the role of yoga and other spiritual practices in the mental health of people [20]. The practice of yoga is most commonly perceived as physical exercise that helps in gaining flexibility, physical strength and helps to relax. In Indian philosophical texts, yoga is treated as a spiritual practice that is related to training of the mind. Patanjali yoga sutra describes yoga as a practice of *'Chitta vritti nirodha'* literally translating as controlling or calming of the mind. Most commonly, a typical yoga schedule follows a combination of *asanas* (postures), *pranayama* (breath control), *pratyahara* (withdrawal of senses), *Dharana* (concentration), *dhyana* (meditation) and *Samadhi* (absorption). While *asanas* are reported to help in improving the physical strength and flexibility, it is argued to also help in building concentration [21]. Preliminary research suggests that *pranayama* calms the nervous system and helps in regulating the blood pressure [22]which is further argued to improve the stress response. *Pratyahara*, *Dharana*, *Dhyana* includes techniques such as; *mantra* chanting,

*yoga nidra*, and *antar mouna* that are said to help in developing an ability to internally witness the sensory inputs [23]. This witnessing capacity is speculated to help one in the reappraisal of the problem in hand, control the fluctuations of the mind and reduce the unconscious negative mental perceptions. Apart from yoga, there are some other spiritual practices such as listening to *satsang* [24], *swadhyaya* (reading Holy Scriptures) [25] and rendering *seva* (selfless service to the community) that have been reported to help maintain wellbeing, reduce stress, anxiety and depression [26, 27].

## Materials and methods

In this research three groups; yoga practitioners, other spiritual practices and non-practitioners were examined for the differences in the measures of illness perception and wellbeing. Additionally, differences based on the duration of practice were also examined in three categories; long-term, mid- term and beginner group. A cross-sectional study was designed using standardized scales for wellbeing related measures and illness perception, questions about daily practices and demographics. An online questionnaire booklet was prepared using google forms and data was collected via social networking groups and email mailing lists. The responses were analyzed using SPSS ver. 26.0.

### Participants

The sample consisted of a total of 668 adults, out of which 96.26% (n = 643) chose to participate and 3.74% (n = 25) declined. Out of the 643 participants, there were 64.7% males (n = 416), 34.7%, females (n = 223), and four preferred not to say about their gender. The age range was 18–72 years (M = 28.12 years, SD = 9.09 years). In the sample, a total of 59% (n = 380) were students and 41% (n = 263) were from non- student groups. The non-student group had 34.4% (n = 221) working adults and the remaining 6.5% (n = 42) were from other categories (retired, homemaker). Concerning qualification, 53.2% (n = 342) participants were from postgraduate and higher qualification, 20.8% (n = 134) from graduate, 23.3% (n = 150) from intermediate or pursuing graduate, and the remaining 2.6% (n = 17) were from high school or below education level. Majority (73.2%) of the participants were from urban (n = 471), whereas 16.5% (n = 106) were from semi-urban, and 10.26% (n = 66) reported being from rural areas of residence.

Within the study sample, 59.6% (n = 384) reported that they practice yoga (includes asana, pranayama, meditation, mantra chanting or any other form of yogic practice) and were categorized as yoga practitioners and 40.4% (259) responded not following any yogic practice. Out of the participants that reported not following any yogic practice, 17.6% (n = 113) reported following one or more of the other forms of spiritual practices for example; watching online spiritual programs (50%), online satsang (14.37%), reading holy scriptures(23.25%), performing seva (12.5%), and were categorized as other spiritual practitioner group. The remaining 22.7% (n = 146) participants reported that they do not follow yogic or any other spiritual practices and they were termed as a non-practitioner group. Further, among the yoga practitioners, 35% (n = 134) were beginners (those who had started yoga practice during COVID19 lockdown period), 39.7% (n = 152) were mid- term (1≤year of practice≤4), and 25.32% (n = 97) were Long term ($\geq$ 5 years of practice) practitioners. Within the beginners, 39.9% reported practicing yoga for all days in the week, 23.9% for 5–6 days, 23.2% for 3–4 days and 13% for 1–2 days in a week. For mid-term practitioners 32.4% reported practicing yoga for all days in the week, 30.4% for 5–6 days, 29.7% for 3–4 days and 7.4% for 1–2 days in a week. For long term practitioners, 58.8% reported practicing yoga for all days in the week, 19.6% for 5–6 days, 11.3% for 3–4 days and 10.3% for 1–2 days in a week. The data for frequency in terms of hours of practice

per day was asked as an open choice question,' How many minutes in a day do you practice yoga? Beginners reported on an average spending 31.24 mins, mid-term practitioner average of 39.10mins, and long term practitioners reported spending average of 51.25 mins, for their daily yoga practice.

## Materials

**Brief Illness Perception.** (BIPQ) [3] was used to measure the individual's perception of COVID19 by adapting the Brief Illness Perception Questionnaire. The adapted version of BIPQ had 12 items designed to rapidly assess the cognitive and emotional representations of COVID19 illness. The five dimensions of cognitive representation of COVID19 illness was assessed through 9 items; identity—the label the person uses to describe the illness and the symptoms they view as being part of the disease (sample item: How much do you think the infected person experiences symptoms from this illness?); consequences—the expected effects and outcome of the illness (sample item: How much does this illness affect the person who suffers from it?); understanding—personal understanding about the cause of the illness (sample item: How well do you feel you understand this illness?); timeline—how long the patient believes the illness will last (sample item: How long do you think the illness last for those who have it?); and cure or control—the extent to which the patient believes that they can recover from or control the illness (sample item: How much control do you feel you have over this illness?). The emotional representation of COVID19 was assessed by 2 items incorporating negative reactions such as fear, anger, and distress (sample item: How much does thinking about this illness affect you emotionally? e.g. does it make you angry, scared, upset or depressed?). Assessment of the causal representation is by an open-ended response item adapted from the IPQ-R, which asks patients to list the three most important causal factors in rank order. All of the items except the causal question are rated using a 0-to10 response scale. The higher the score is, the greater the perception of the illness for that particular item. The total scale alpha coefficient in this study sample was 0.64.

**Depression, Anxiety and Stress Scale.** (DASS-9) [28] was used to measure the depression, anxiety and stress experienced by the participants during the COVID19 lockdown period using DASS-9. It is the shorter version of DASS-42 [29] and consists of three sub-factors with 3 items each viz., depression (sample item: I found it difficult to work up the initiative to do things), anxiety (sample item: I experienced trembling eg. in the hands), and stress (sample item: I tended to overreact to). The instructions were modified to suit the current research and participants were asked to rate how much each statement applied to them during the lockdown period. Cronbach's alpha for the total DASS-9 was reported by Yusoff (2013) equal to .72 whereas for Depression, Anxiety and Stress factors, alphas were .52, .57, and .55, respectively [29]. In this study sample the Cronbach alpha for total DASS-9, depression, anxiety and stress was found to be 0.73, 0.63, 0.64, and 0.53 respectively.

**Scale of General Wellbeing.** (SGWB) [30] was used to measure the general wellbeing through fourteen common constructs as indicators of well-being viz., happiness, vitality, calmness, optimism, involvement, self-awareness, self-acceptance, self-worth, competence, development, purpose, significance, self-congruence and connection (sample item: I accept most aspects of myself). All items were phrased positively and rated on a Likert scale from 1 (Not at all) to 5 (very true), indicating experiences in life overall. Previous studies have reported a Cronbach alpha for SGWB ranging from 0.82 to 0.92 [30]. In this study sample, the Cronbach alpha for the total scale was found to be 0.93.

**Brief Resilience Scale.** (BRS) [31] was used to measure resilience. The scale contains 6 items measuring the ability to bounce back from stress and difficulties (e.g., "I usually come

through difficult times with little trouble"). The items are rated on a 5-point Likert scale from 1 (Strongly Disagree) to 5 (Strongly Agree). The possible score ranges from 1 (minimum resilience) to 6 (maximum resilience). Three items are negatively worded and are reversed scored. Adequate reliability, with Cronbach's alpha ranging from .80 to .91 in 4 different samples was reported in an earlier study [31]. In this study sample, the Cronbach alpha was found to be 0.73.

**Peace of Mind Scale.** (POMS) [15] was used to measure peace of mind through a single factor model presented by POMS. The scale consists of a 5 item scale that measures affective wellbeing. The items reflect the experiences of internal peace and harmony in general terms (e.g., I have peace and harmony in my mind) as well as in everyday circumstances (e.g., I feel content and comfortable with myself in daily life). Participants indicated how often they experience the internal states described in each of the items on a scale of 1 (not at all) to 5 (all the time). The five-item POMS (Cronbach alpha = 0.78) was used in this study which had previously been confirmed for the Indian population [32]. In this study sample, the Cronbach alpha was found to be 0.91.

**Emotion Regulation Questionnaire.** (ERQ) [33] was used to assess the commonly used strategies to alter emotion through ERQ viz., 6 items on cognitive reappraisal (sample items: "When I'm faced with a stressful situation, I make myself think about it in a way that helps me stay calm"), and 4 items on expressive suppression (sample items: "When I am feeling negative emotions, I make sure not to express them"). Participants responded to each item using a 7-point Likert scale ranging from 1 (strongly disagree) to 7 (strongly agree). The average of all the scores in each subscale of cognitive reappraisal and expressive suppression are used for analysis. The higher the score represents greater the use of that particular emotion regulation strategy, conversely lower scores means less frequent use. In a four study sample reported the Cronbach alpha ranging for reappraisal facet 0.75 to 0.82 and suppression factor 0.68–0.76 [34]. In this study sample, the Cronbach alpha for cognitive reappraisal and expressive suppression was found to be 0.83 and 0.75 respectively.

Apart from the standardized scales as described above, the data collection booklet included consent form, yoga schedule, and demographic information schedule. The categorization of the yoga practitioners, and non-practitioners was based on the dichotomous question; 'Do you practice yoga (includes asana, pranayama, meditation, mantra chanting or any other form of yogic practice) in your daily routine?" The non-yoga practitioner group was further classified based on multiple response question; 'Any other form of spiritual practice do you follow?' for example; online *Satsang* (listening to devotional songs), watching spiritual programs, reading Holy Scriptures, selfless service or any other. This formed a group of other spiritual practitioners and the rest of non-yoga practitioners formed a third group classified as non-practitioners. The yoga practitioners were also asked about the duration of their practice. The demographic profile consisted of information about age, gender, qualification, working status and place of residence. An additional item on the three most important causal factors (in rank order) of stress during lockdown was also asked through a question 'Please list in rank-order the THREE most important factors that you believe are reasons for stress due to lockdown'.

## Procedure

**Preparation for the study.** The study was designed for both Hindi and English speaking population keeping in mind the diversity in the preference for language in the population. At the outset, the original English scales for which the Hindi version was not available (ERQ and BRS), were translated into Hindi by a bilingual expert working in the area of psychological assessment. The Hindi translations of all the scales were then evaluated by the first and second

author to check for adequacy of translation. Modifications were made wherever the Hindi translations were not found to adequately capture the intended meaning. Further, a bilingual expert independently back-translated these scales from Hindi to English. The back translations were again reviewed by the first and second author and matched to the original scales. At this stage, most items were found to aptly represent the content of the original English scales. The finalized Hindi and English scales were used to prepare the data collection booklet.

### Data collection and analysis

The cross-sectional study was conducted using an online survey. The sample for the study was recruited through the distribution of an online message consisting a brief introduction to the study and a link to the google form of data collection booklet using social networks, mailing lists and snowballing techniques. The online message with the link was especially circulated among the yoga practitioner groups. An electronic consent was obtained from each of the participants before beginning the survey. The data collected was anonymous and no personal details that could identify the participants were asked in the google form. Participants were assured that the data will be kept confidential and only be used for research purposes. The google form was available for responses from 26th April -8th June 2020 (beginning of Unlock 1.0), corresponding to four to ten weeks of the lockdown in India.

In the first step of data analysis, the responses for the Hindi (n = 96) and English (n = 547) scale were analyzed using an independent t-test that showed no significant difference in illness perception or wellbeing related measures in any of the practitioner groups due to difference in language (all ps>0.05). Therefore, the remaining analysis was conducted on the combined English and Hindi data. In the next step, the descriptive analysis was conducted and the internal reliability scores (Cronbach alpha) for each scale was computed. To confirm the factor structure of the scales used for this study sample, the data were subjected to confirmatory factor analysis. Multivariate analysis (MANOVA) was conducted to examine the differences in the mean scores of illness perception and wellbeing related measures among the yoga practitioners, other spiritual practitioners, and non- practitioner groups. The open-ended question was analyzed using percentage analysis. Lastly, MANOVA was also performed within the yoga practitioner group based on the duration of practice.

## Results

### Descriptive analysis

The descriptive statistics of all the dependent variables were analyzed. Three outliers identified based on extreme values more than three IQR's (interquartile range) [35] computed from Tukey's hinges in SPSS, were deleted. Confirmatory factor analysis (CFA) showed that most of the fit statistics for all the scales were in the acceptable range. Statistics presented in Table 1.

### Demographic variables

The relationship between demographic variables (age, gender, qualification, working status, and place of residence) was examined on illness perception (BIPQ), wellbeing related measures (DASS, SGWB, POMS, BRS) and Emotion regulation strategies (ERQ). MANOVA results indicated no statistically significant main effect of working status and qualification on the COVID19 perception, wellbeing measures (DASS, SGWB, POMS, and BRS) or emotion regulation strategies (ERQ). However, there was a significant effect of gender on the mean scores of illness concern $F(1,634) = 11.14$, p<0.001, partial eta squared = .02, and emotional representation of COVID19 $F(1,634) = 10.50$, p<0.001, partial eta squared = .02, partial eta squared

**Table 1. Goodness of fit statistics for confirmatory factor analysis.**

| Measures of goodness fit | Acceptable level | BIPQ | DASS | POMS | SGWB | BRS | ERQ |
|---|---|---|---|---|---|---|---|
| χ2 (df) | | 118.75 (26) | 86.78 (24) | 16.66 (3) | 281.53 (70) | 32.09 (5) | 202.38 (34) |
| χ2 /df | <5 [36] | 4.57 | 3.61 | 5.55 | 4.02 | 6.41 | 5.95 |
| p-value | >0.05 | 0.000 | 0.000 | 0.001 | 0.000 | 0.000 | 0.000 |
| RMSEA | <0.10 [37] | 0.077 | 0.065 | 0.084 | 0.062 | 0.093 | 0.092 |
| CFI | ≥0.95 [38] | 0.915 | 0.953 | 0.995 | 0.967 | 0.970 | 0.927 |
| GFI | >0.90 [39] | 0.961 | 0.971 | 0.990 | 0.941 | 0.982 | 0.941 |
| AGFI | >0.90 [39] | 0.932 | 0.945 | 0.949 | 0.912 | 0.926 | 0.904 |

Notes: CFI–comparative fit index; AGFI–adjusted goodness of fit index; GFI–goodness of fit index; RMSEA–root mean square estimation. BIPQ-Brief Illness Perception Questionnaire(adapted to COVID19); DASS-Depression, Anxiety and Stress Scale; POMS-Peace of Mind Scale; SGWB-Scale of General Wellbeing; BRS-Brief Resilience Scale; ERQ-Emotion regulation Questionnaire.

= .02, with females reporting higher illness concern (M = 8.18, SD = 2.20),and higher emotional impact of COVID19 (M = 5.73, SD = 3.03) than males illness concern (M = 7.50, SD = 2.83), and emotional impact of COVID19 (M = 5.00, SD = 3.03), respectively. There was a significant effect place of residence on illness consequence of COVID19 $F$ (1,634) = 5.61, p<0.05, partial eta squared = -.01, with urban participants reporting higher concern for consequences of COVID19 (M = 7.76, SD = 5.34) than semi-urban (M = 6.95, SD = 2.55) or rural (M = 7.22, SD = 2.88).

Age had a significant effect on depression (DASS-D) $F$ (1,634) = 9.34, p<0.005, partial eta squared = -.01 and Peace of Mind (POMS) $F$ (1,634) = 13.02, p<0.001, partial eta squared = -.02, with participants from age group 18–25 years reporting higher depression (DASS-D) (M = 0.97, SD = 0.70) than age group 26–35 years (M = 0.81, SD = .60), age group 36–45 years (M = 0.64, SD = 0.62) and age group 46 and above (M = 0.61, SD = 0.57). Lower mean scores for Peace of Mind (POMS) were reported by the participants of age group 18–25 years (M = 3.09, SD = 0.94) than age group 26–35 years (M = 3.42, SD = 1.07), age group 36–45 years (M = 3.55, SD = 1.07) and age group 46 and above (M = 3.80, SD = 1.00).

## Effect of yoga and other spiritual practice on illness perception, and wellbeing measures

Before conducting the MANOVA, Pearson correlation was performed between all dependent variables to test the multivariate assumption that the dependent variables would be correlated with each other in the moderate range [40]. A meaningful pattern of correlations was observed amongst most of the dependent variables (r = - 0.460 to r = 0.448), suggesting the appropriateness of MANOVA. Correlations are presented in S1 Table. Additionally, the BOX's M value of 575.82 was associated with less than a p-value of 0.001, which was interpreted as non-significant based on Huberty and Petoskey's (2000) guidelines (i.e p < .005) [41]. Thus, the covariance matrices between the groups were assumed to be equal for the MANOVA.

The MANOVA was conducted to test the hypothesis that there would be one or more mean differences between the spiritual practitioner levels (yoga practitioners, other spiritual practitioners, and non-practitioners) and COVID19 perception, wellbeing related measures, as well as in their emotion regulation strategies. After controlling for the confounding effect of demographic variables a statistically significant MANOVA effect was obtained, Pillai's' Trace = .19, $F$ (38, 1220) = 3.13, p<0.001. The multivariate effect size was estimated at .156.

A series of Levene's $F$ tests, to examine the homogeneity of variance assumption was conducted and the statistics are presented in Table 2. In the third stage, a series of post hoc

**Table 2. One way ANOVA with illness perception (IP), wellbeing measures, and emotion regulation (ERQ) as dependent variable and practitioner groups as the independent variable.**

| | Levene's statistics | | ANOVA | | |
| --- | --- | --- | --- | --- | --- |
| | F | Sig. | F | Sig. | Partial Eta Squared |
| Consequence(IP1) | 2.440 | .088 | .677 | .508 | .002 |
| Timeline (IP2) | 3.594 | .028 | 1.349 | .260 | .004 |
| Personal control (IP3) | 1.377 | .253 | 16.971 | .000 | .051 |
| Treatment Control (IP4) | .126 | .881 | 2.877 | .057 | .009 |
| Identity (IP5) | 1.216 | .297 | .850 | .428 | .003 |
| Illness concern(IP6) | 2.831 | .060 | 1.166 | .312 | .004 |
| Coherence/understanding (IP7) | .844 | .431 | 5.884 | .003 | .018 |
| Emotional representation (IP8) | 4.173 | .016 | 4.413 | .012 | .014 |
| Risk perception (IP10) | .598 | .550 | 3.724 | .025 | .012 |
| Risk perception (IP11) | .021 | .980 | 4.460 | .012 | .014 |
| Personal preventative control(IP12) | .976 | .378 | 7.532 | .001 | .023 |
| Depression (DASS-D) | 6.441 | .002 | 19.463 | .000 | .058 |
| Anxiety (DASS-A) | .136 | .873 | .355 | .701 | .001 |
| Stress (DASS-S) | 1.103 | .333 | 4.955 | .007 | .015 |
| Peace of Mind (POMS) | .129 | .879 | 40.851 | .000 | .114 |
| Wellbeing (SGWB) | 2.880 | .057 | 40.271 | .000 | .112 |
| Resilience(BRS) | .548 | .578 | 1.696 | .184 | .005 |
| Emotion regulation-Cognitive Appraisal (ERQ-C) | .731 | .482 | 19.970 | .000 | .059 |
| Emotion regulation-Expressive Suppression (ERQ-E) | .174 | .841 | .419 | .658 | .001 |

analyses (Tukey's HSD) were performed to examine the individual mean difference comparisons across all three groups. The summary of post hoc comparisons is presented in Table 3.

**COVID19 perception-BIPQ.** There was a statistically significant difference in group means for the personal control (IP3) $F(2,639) = 14.81$, $p < .001$, Coherence/ understanding (IP7) $F(2,639) = 4.95$, $p < .01$ Emotional representation (IP8) $F(2,639) = 5.17$, $p < .0.01$, Risk perception (IP10 and IP11) $F(2,639) = 4.01$, $p < .01$, Personal preventive Control (IP12) $F(2,639) = 6.30$, $p < .01$. However, there were no significant differences found in the COVID19 representation of illness perception with respect to consequence (IP1), timeline (IP2), treatment control (IP4), identity (IP5) and illness concern (IP6).

A post hoc analysis (Tukey HSD) revealed significant differences in mean scores of yoga practitioner group for personal control (IP3)(M = 6.22, SD = 2.72), coherence/understanding (IP7) (M = 7.58, SD = 2.22), and Emotional representation (IP8) (M = 4.95, SD = 3.20) when compared to the mean scores of other spiritual practitioner group for personal control (M = 5.06, SD = 2.62), coherence/understanding (IP7)(M = 7.29, SD = 2.44), emotional representation (IP8)(M = 5.74, SD = 2.87), and also for mean scores of non-practitioner for personal control (M = 4.88, SD = 2.6), Coherence/understanding (IP7) (M = 6.83, SD = 2.28), emotional representation (IP8)(M = 5.66, SD = 2.97), all ps<0.05 with higher mean interpreted as a higher perception of personal control over the illness /COVID19, higher coherence/understanding and higher emotional representation of COVID19. However, there was no significant difference between the other spiritual practices and the non-practitioner group, all ps >0.05.

There was also a statistically significant difference in the mean scores of yoga practitioner for risk perception (IP10 and IP11) (M = 5.88, SD = 2.75), when compared with the mean scores of non-practitioner risk perception (M = 6.59, SD = 2.58), p<0.05, with lower mean scores interpreted as a lower perception of risk to contract COVID19. However, there was no

**Table 3. Summary of the post hoc comparison for the three groups- yoga practitioners, other spiritual practitioner and non- practitioner groups.**

| Measures | Yoga Vs other Spiritual practitioner group | Other spiritual vs non-practitioner group | Yoga vs non-practitioner group |
|---|---|---|---|
| Consequence(IP1) | n.s. | n.s. | n.s. |
| Timeline (IP2) | n.s. | n.s. | n.s. |
| Personal control (IP3) | p<0.05 | n.s. | p<0.05 |
| Treatment Control (IP4) | n.s. | n.s. | n.s. |
| Identity (IP5) | n.s. | n.s. | n.s. |
| illness concern(IP6) | n.s. | n.s. | n.s. |
| coherence/understanding (IP7) | p<0.05 | n.s. | p<0.05 |
| Emotional representation (IP8) | p<0.05 | n.s. | p<0.05 |
| Risk perception (IP10) | p<0.05 | n.s. | p<0.05 |
| Peer Risk perception (IP11) | p<0.05 | n.s. | p<0.05 |
| Personal preventative control(IP12) | p<0.001 | n.s. | p<0.001 |
| Depression (DASS-D) | p<0.001 | n.s. | p<0.001 |
| Anxiety (DASS-A) | p<0.05 | n.s. | p<0.05 |
| Stress (DASS-S) | p<0.001 | p<0.05 | p<0.001 |
| Peace of Mind (POMS) | p<0.05 | p<0.05 | p<0.05 |
| Wellbeing (SGWB) | p<0.05 | n.s. | p<0.05 |
| Resilience(BRS) | n.s. | n.s. | n.s. |
| Emotional regulation-Cognitive Appraisal (ERQ-C) | p<0.05 | p<0.05 | p<0.05 |
| Emotional regulation-(Expressive Suppression (ERQ-E) | n.s. | n.s. | n.s. |

Note: n.s.-not significant.

significant difference in the mean scores of other spiritual practitioner group when compared with the yoga practitioner group, and the non-practitioner group, both ps>0.05.

There was also a statistically significant difference in the mean scores for the perception of preventive control (IP12) of yoga practitioner (M = 7.10, SD = 2.44), when compared with other spiritual practitioner group (M = 6.75, SD = 2.23), and non-practitioner (M = 6.17, SD = 2.62), p<0.001, with higher mean scores interpreted as a higher perception of personal preventive control over COVID19. However, there was no significant difference in the mean scores of the other spiritual practitioner group, when compared with the yoga practitioner group and the non-practitioner group, both ps>0.05. Means plot is shown in Fig 1.

Percentage analysis of the causal representation of COVID19 on the responses received to the open-ended question (IP9) which asks patients to list the three most important causal factors in the rank order revealed that 48.25% ranked lack of adequate and timely information as the foremost reason for the spread of COVID19, followed by consumption of animal flesh (17.06%) and international movement of tourists and immigrants (16.74%). Other causal factors listed by participants included; lack of immunity, lack of medical facilities/treatment, improper/late action on lockdown measures etc.

**Wellbeing related measure (DASS, SGWB, POMS, BRS).** There was a statistically significant difference in group means for depression (DASS-D) ($F$ (2,639) = 12.48, $p < .001$, partial eta square = .058, and stress (DASS-S) ($F$ (2,639) = 3.80, $p < .05$, partial eta square = .015. However, there was no statistically significant difference in group means for anxiety (DASS-A), $p>.05$.

Further, a post hoc analysis (Tukey HSD) revealed significant differences in mean scores for depression (DASS-D) among yoga practitioner group (M = 0.74, SD = .63), when

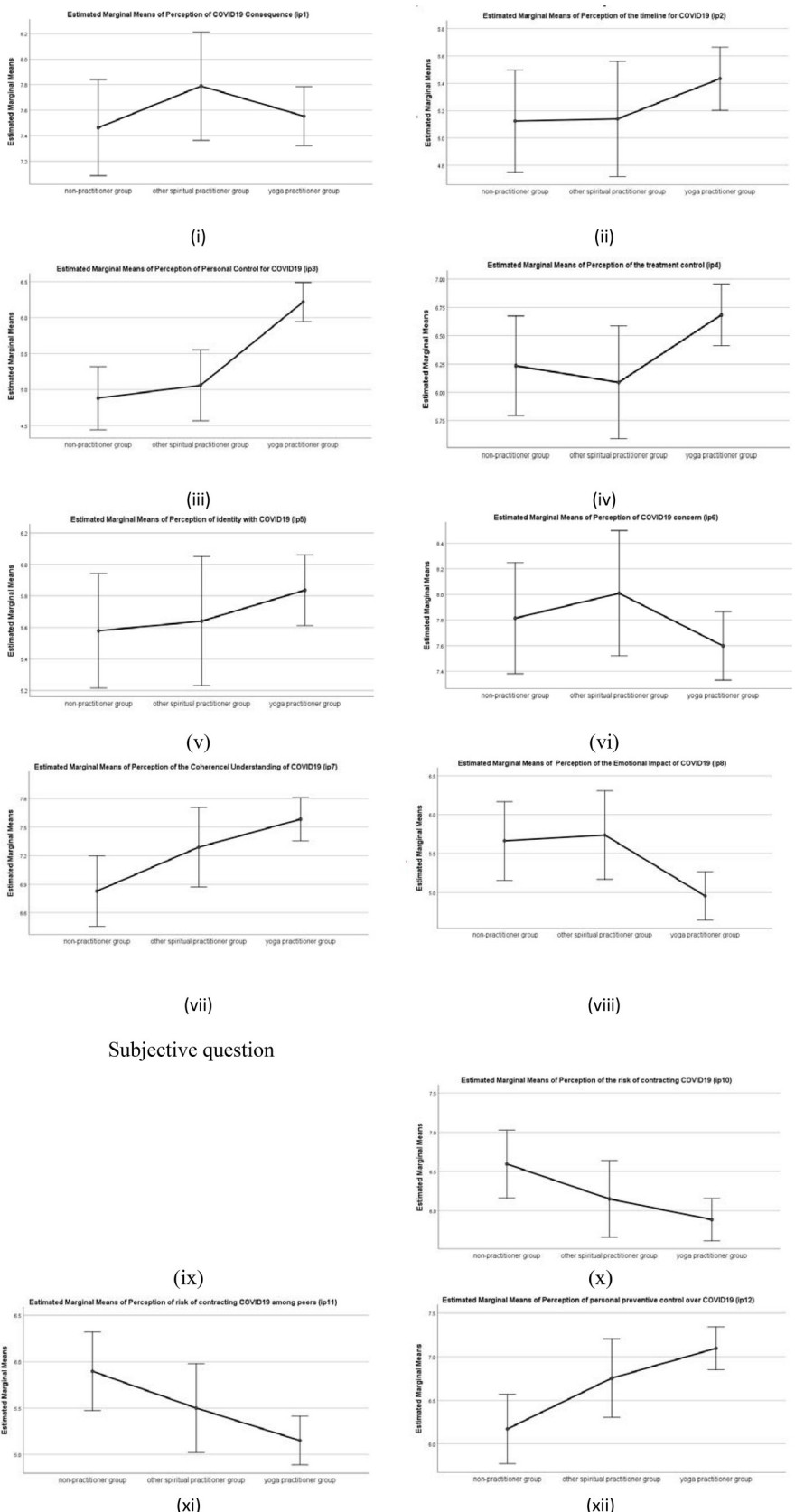

Subjective question

**Fig 1.** Means Plot for COVID19 perception of (i) consequence; (ii) timeline; (iii) personal control; (iv) treatment control; (v) identity; (vi) illness concern; (vii) coherence/ understanding; (viii) emotional representation; (x) risk perception; (xi) peer risk perception; (xii) preventive control for three practitioner groups. Note: Error bars at 95% Cl.

compared with other spiritual practitioner group (M = .93, SD = .58), and non-practitioner group (M = 1.13, SD = .73), both ps< 0.001, with higher mean scores signifying higher depression. There was also a significant difference in mean scores for stress (DASS-S) for yoga practitioner group (M = .79, SD = .61) when compared with other spiritual practitioner group (M = .94, SD = .60), and non-practitioner group (M = .95, SD = .61), both ps< 0.001, with higher mean scores signifying higher stress. There was also a significant difference in the mean scores for stress (DASS-S) between other spiritual practitioners and the non- practitioner, p<0.05. Whereas, there was no statistically significant difference for anxiety or depression between the other spiritual practitioner group and non-practitioner group p>0.05.

Percentage analysis of the additional item on the three most important causal factors (in rank order) of stress during lockdown revealed that majority of the participants reported isolation due to lockdown (23.17%), fear of loss of job/business (financial insecurity) (18.57%), and fear of contracting virus (10.32%) as three main causes of stress during COVID19 lockdown. Other reasons of stress included; media reports and inadequate information (7.93%), the uncertainty of future (7.40%), routine disturbances (6.66%), educational loss (5.87%), and family issues (3.65%). Interestingly, 8.89% of the participants reported having no stress at all. A gender wise analysis of the causal factors of stress shows that females higher percentage of females (25.89%) reported stress due to isolation due to lockdown than males (22.11%). Whereas, financial insecurity was ranked as major cause of stress by more males (19.59%) than females (17.41%).

There was also a statistically significant difference in group means for wellbeing (SGWB) $F$ (2,639) = 31.20, $p < .001$, partial eta squared = .112, and peace of mind (POMS) $F$ (2,639) = 30.99, $p < .001$, partial eta square = .114. However, there was no statistically significant difference in group means for resilience (BRS), $p < .05$. Further, a post hoc analysis (Tukey HSD) revealed that mean scores of yoga practitioner group for wellbeing (SGWB) (M = 3.74, SD = .78), and peace of mind (POMS) (M = 3.565, SD = .96) differed significantly when compared with the other spiritual practitioner group mean scores for wellbeing (SGWB) (M = 3.28, SD = .67), peace of mind (POMS) (M = 3.13, SD = .96), and non-practitioner group mean scores for wellbeing (SWGB) (M = 3.11, SD = .85), peace of mind (POMS) (M = 2.73, SD = .99), all ps< 0.001. There was no significant difference between the mean scores for wellbeing (SWGB) between other spiritual practitioner groups and non-practitioner groups p>0.05. However, for peace of mind (POMS) there was a statistically significant difference between the other spiritual practitioner group and non-practitioner group p<0.001, with higher mean scores signifying higher wellbeing and higher peace of mind. Means plot is shown in Fig 2.

**Emotion regulation strategies (ERQ).** There was a statistically significant difference in group means for cognitive reappraisal strategies (ERQ-C) $F$ (2,639) = 14.85, $p < .001$, partial eta square = .059. However, there was no significant difference in the group mean scores for Expressive suppression (ERQ-E), p>0.05. Further, a post hoc analysis (Tukey HSD) revealed significant differences in mean scores for yoga practitioner group (M = 5.24, SD = 1.07), other spiritual practitioner group (M = 4.85, SD = 1.17), and non-practitioner group (M = 4.57, SD = 1.19), both ps< 0.001. There was also a statistically significant difference between the other spiritual practitioner group and non-practitioner group p<0.001, with higher mean scores signifying higher emotion regulation through cognitive reappraisal strategies. Means plot is shown in Fig 3.

**Fig 2.** Means plot for Wellbeing measures (i) Depression; (ii) Anxiety; (iii) Stress; (iv) Wellbeing; (v) Peace of Mind and (vi) Resilience for three practitioner groups. Note: Error bars at 95% Cl.

### Effect of duration of yoga practice on illness perception and wellbeing measures

The MANOVA was conducted to test the hypothesis that there would be one or more mean differences in the wellbeing for different groups of yoga practitioners differentiated based on

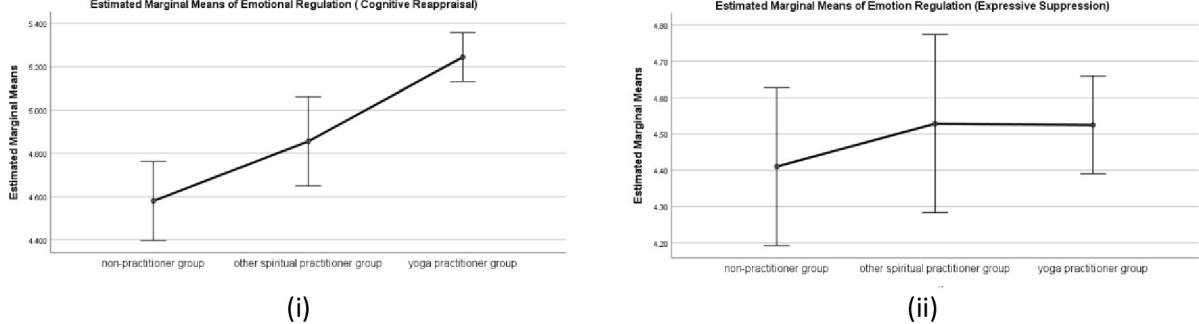

**Fig 3.** Means Plot for Emotion regulation i) Cognitive appraisal ii) Expressive suppression for three practitioner groups. Note: Error bars at 95% Cl.

**Table 4. One-way ANOVA with wellbeing measures as the dependent variable and yoga practitioner groups based on the duration of practice as the independent variable.**

| | Levene's statistics | | ANOVA | | |
|---|---|---|---|---|---|
| | F | Sig. | F | Sig. | Partial eta squared |
| Consequence(IP1) | 9.185 | 0.000 | 2.239 | 0.108 | 0.012 |
| Timeline (IP2) | 2.511 | 0.083 | 0.325 | 0.723 | 0.002 |
| Personal control (IP3) | 0.101 | 0.904 | 3.571 | 0.029 | 0.019 |
| Treatment Control (IP4) | 0.036 | 0.965 | 0.675 | 0.510 | 0.004 |
| Identity (IP5) | 0.172 | 0.842 | 0.292 | 0.747 | 0.002 |
| Illness concern(IP6) | 16.833 | 0.000 | 6.700 | 0.001 | 0.035 |
| Coherence/understanding (IP7) | 0.377 | 0.686 | 1.279 | 0.280 | 0.007 |
| Emotional representation (IP8) | 3.877 | 0.022 | 3.311 | 0.038 | 0.018 |
| Risk perception (IP10) | 4.312 | 0.014 | 1.535 | 0.217 | 0.008 |
| Risk perception (IP11) | 2.614 | 0.075 | 5.640 | 0.004 | 0.030 |
| Personal preventative control(IP12) | 0.359 | 0.699 | 1.428 | 0.241 | 0.008 |
| Depression (DASS-D) | 4.667 | 0.010 | 4.151 | 0.017 | 0.022 |
| Anxiety (DASS-A) | 7.746 | 0.001 | 5.189 | 0.006 | 0.027 |
| Stress(DASS-S) | 0.934 | 0.394 | 2.708 | 0.068 | 0.015 |
| Peace of Mind (POMS) | 1.232 | 0.293 | 15.100 | 0.000 | 0.076 |
| Wellbeing (SGWB) | 9.611 | 0.000 | 22.606 | 0.000 | 0.110 |
| Resilience(BRS) | 4.646 | 0.010 | 1.920 | 0.148 | 0.010 |
| Emotion regulation-Cognitive Appraisal (ERQ-C) | 0.834 | 0.435 | 5.298 | 0.005 | 0.028 |
| Emotion regulation-(Expressive Suppression (ERQ-E) | 1.373 | 0.255 | 2.337 | 0.098 | 0.013 |

the number of practice years. After controlling for the confounding effect of demographic variables a statistically significant MANOVA effect was obtained, Pillai's' Trace = .229, $F$ (38, 716) = 2.43, p<0.001. The multivariate effect size was estimated at .114.

Before conducting a series of follow up ANOVAs, the homogeneity of variance assumption was tested for all the dependent variables. A series of Levene's $F$ tests, to examine the homogeneity of variance assumption was conducted and is presented in Table 4. A series of one-way ANOVA was conducted followed by a series of post hoc analyses (Tukey's HSD) were performed to examine the individual mean difference comparisons across all three groups of spiritual practitioners and all the dependent variables. The results revealed statistically significant comparisons as listed in Table 5.

**COVID19 perception (BIPQ).** There was a statistically significant difference in group means for the Personal control (IP3) $F(2,367)$ = 3.571, $p$ < .05, partial eta squared = 0.02, Illness concern(IP6) $F(2,367)$ = 6.70, $p$ < .001, Emotional representation (IP8) $F(2,367)$ = 3.31, $p$ < .0.05, Risk perception(IP11) $F(2,367)$ = 5.64, $p$ < .005. However, there was no significant difference found in the mean scores of COVID19 representation of illness perception for the consequence (IP1), timeline (IP2), treatment control (IP4), identity (IP5) and Coherence/ Understanding (IP7)

A post hoc analysis (Tukey's HSD) revealed significant differences in mean scores of long term practitioner for personal control (IP3) (M = 6.68, SD = 2.79), illness concern(IP6) (M = 6.93, SD = 3.3), emotional representation (IP8) (M = 4.27, SD = 2.80), risk perception (IP10 and IP11) (M = 5.40, SD = 3.04) when compared with beginners mean scores of personal control (IP3) (M = 5.75, SD = 2.68),illness concern (M = 8.23, SD = 2.11), emotional representation (M = 5.44, SD = 2.89), risk perception(IP10 and IP11) ((M = 6.14, SD = 2.40), all ps<0.005, with higher mean interpreted as a higher perception of personal control, illness

**Table 5. Summary of the post hoc analysis for the groups based on duration of yoga practice- long-term, mid- term, and beginner groups.**

| Measures | Long-term vs mid-term group | Mid-term vs beginner group | Long-term vs beginner group |
|---|---|---|---|
| Consequence(IP1) | n.s. | n.s. | n.s. |
| Timeline (IP2) | n.s. | n.s. | n.s. |
| Personal control (IP3) | n.s | p < .005 | p < .005 |
| Treatment Control (IP4) | n.s. | n.s. | n.s. |
| Identity (IP5) | n.s. | n.s. | n.s. |
| illness concern(IP6) | n.s. | p < .005 | p < .005 |
| coherence/understanding (IP7) | n.s. | n.s. | n.s. |
| Emotional representation (IP8) | n.s. | p < .005 | p < .005 |
| Risk perception (IP10) | n.s. | p < .005 | p < .005 |
| Peer Risk perception (IP11) | n.s. | p < .005 | p < .005 |
| Personal preventative control(IP12) | n.s. | p < .005 | p < .005 |
| Depression (DASS-D) | p<0.001 | n.s. | p<0.001 |
| Anxiety (DASS-A) | p<0.05 | n.s. | p<0.05 |
| Stress (DASS-S) | n.s. | n.s. | n.s. |
| Peace of Mind (POMS) | p<0.001 | p<0.001 | p<0.001 |
| Wellbeing (SGWB) | p<0.05 | n.s. | p<0.05 |
| Resilience(BRS) | n.s. | n.s. | n.s. |
| Emotional regulation-Cognitive Appraisal (ERQ-C) | | n.s. | p<0.05 |
| Emotional regulation-(Expressive Suppression (ERQ-E) | n.s. | n.s. | n.s. |

Note: n.s.-not significant.

concern, emotional impact and higher risk perception of contracting COVID19. There was also a statistically significant difference in mean scores between beginners' illness concern (M = 8.23, SD = 2.11), and mid-term (M = 7.53, SD = 2.67) p<0.005. However, there was no significant difference between the long term and mid-term practitioners group for illness concern p>0.05. Means Plot shown in Fig 4.

**Wellbeing related measures (DASS, SGWB, POMS, BRS).** There was a statistically significant difference in group means for depression (DASS-D) ($F$ (2,367) = 4.15, $p < .05$, partial eta square = .022, anxiety (DASS-A) $F$ (2,367) = 5.19, $p < .01$, partial eta square = .03 and a trend in group means for stress (DASS-S), $F$ (2,367) = 2.71, $p$ = .068. Further, a post hoc analysis (Tukey HSD) revealed significant differences in mean scores between long term practitioner group depression (DASS-D)(M = 0.55, SD = .55), Anxiety (DASS)(M = .26, SD = .38), and mid-term practitioner group depression (DASS-D) (M = .81, SD = .67), anxiety (DASS-A) (M = .48, SD = .56), and beginners group depression (DASS-D) (M = .80, SD = .60), all ps< 0.005, with higher mean scores signifying higher depression, and anxiety. There was no statistically significant difference between the mid-term and beginner practitioner group.

There was a statistically significant difference in group means for SGWB ($F$ (2,367) = 22.60, $p < .001$, partial eta square = .110, and POMS ($F$ (2,375) = 15.10, $p < .001$, partial eta square = .076. However, there was no statistically significant difference found in group means for BRS, $p < .05$. Post hoc analysis (Tukey HSD) revealed significant differences in mean scores of Wellbeing (SGWB) of long term practitioner group (M = 4.10, SD = .64) when compared with mid- term practitioner group (M = 3.79, SD = .65) and beginners group (M = 3.42, SD = .84), both ps< 0.001. Mean scores of POMS for long-term practitioner group (M = 4.03, SD = .91) also differed significantly with the mean scores of mid-term practitioner group (M = 3.50, SD = .87), and beginner practitioner group (M = 3.27, SD = .94), both ps< 0.001. There was also a statistically significant difference in the wellbeing (SGWB) mean scores and peace of mind

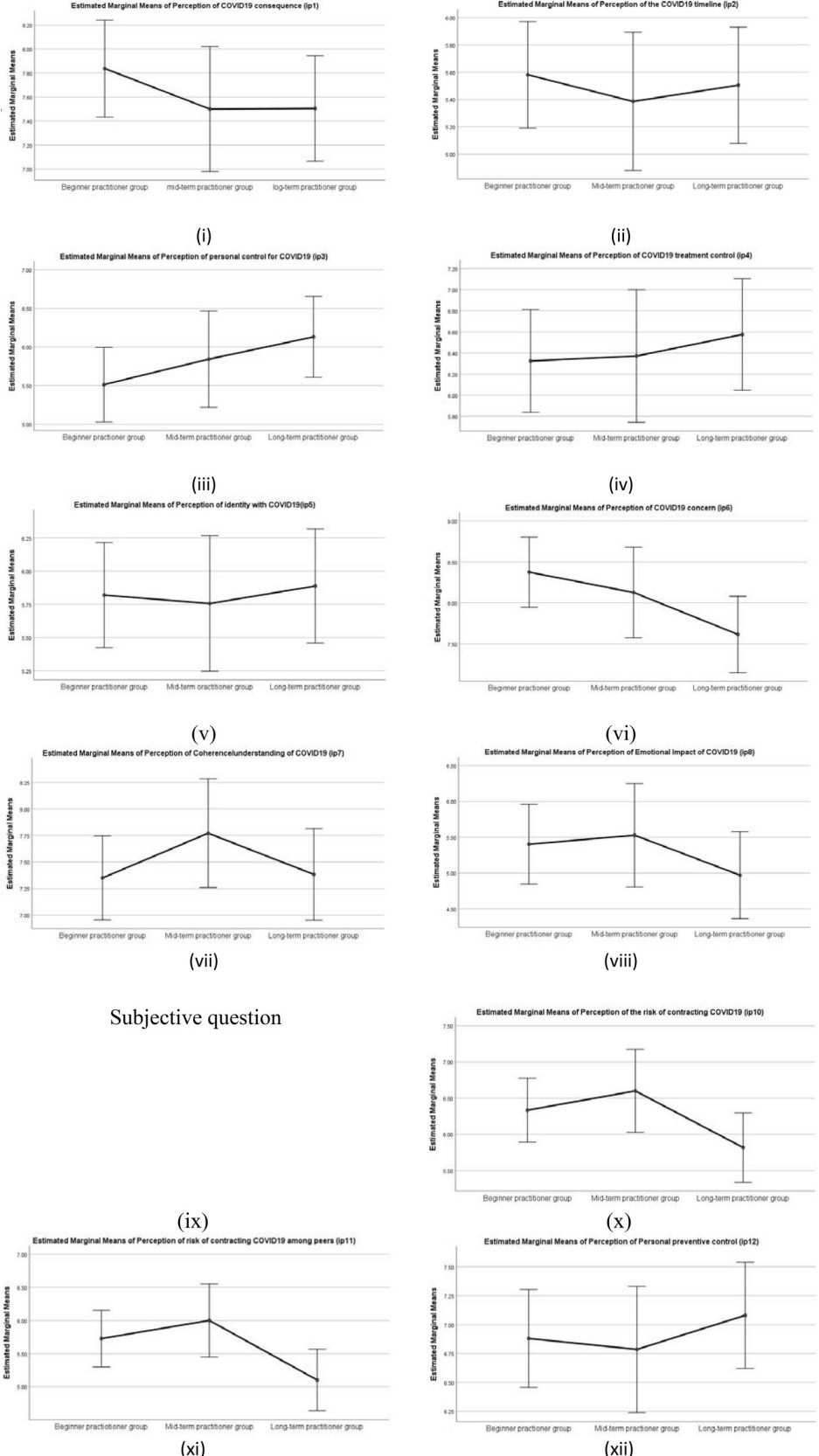

Subjective question

**Fig 4.** Means Plot for COVID19 perception of (i) consequence; (ii) timeline; (iii) personal control; (iv) treatment control; (v) identity; (vi) illness concern; (vii) coherence/ understanding; (viii) emotional representation; (x) risk perception; (xi) peer risk perception; (xii) preventive control for three yoga practitioner groups based on duration of the yoga practice. Note: Error bars at 95% Cl.

(POMS) mean scores between the mid-term practitioner group and beginner group p>0.05, with higher mean scores signifying higher peace of mind. Means plot is shown in Fig 5.

**Emotion regulation measure (ERQ).** There was a statistically significant difference in group means for ERQ (cognitive reappraisal) ($F$ (2,375) = 5.30, $p$ < .005, partial eta square = .028. However, there was no significant difference in the group mean scores for ERQ (Expressive suppression), p>0.05), indicating that the duration of yoga practice affects cognitive reappraisal strategies for emotion regulation. Further, a post hoc analysis (Tukey HSD) revealed significant differences in mean scores between long-term practitioner group (M = 5.51, SD = .97), and beginner group (M = 5.0, SD = 1.02) with higher mean scores signifying higher

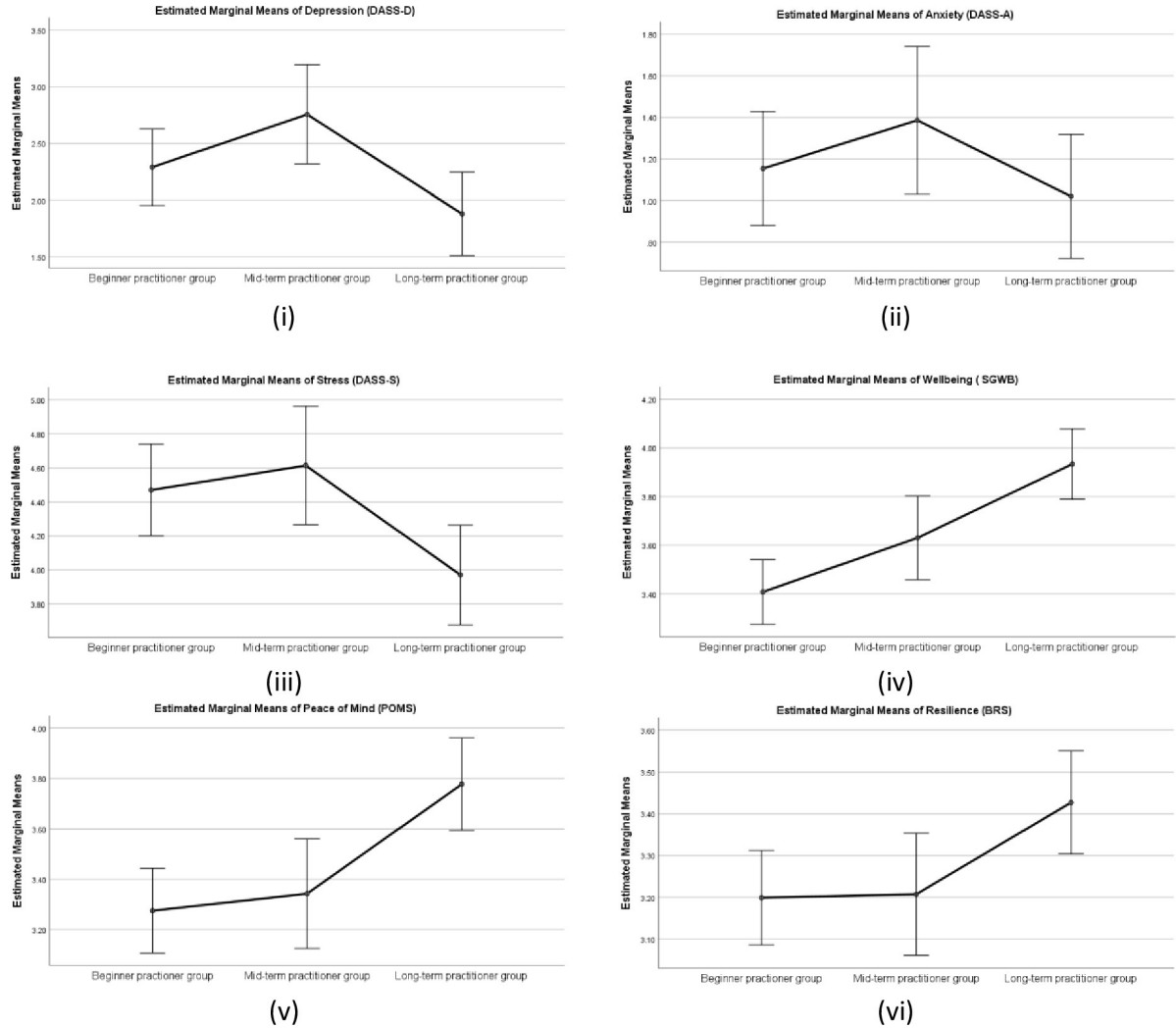

**Fig 5.** Means plot for Wellbeing measures (i) Depression; (ii) Anxiety; (iii) Stress; (iv) Wellbeing; (v) Peace of Mind and (vi) Resilience for three yoga practitioner groups based on duration of the yoga practice. Note: Error bars at 95% Cl.

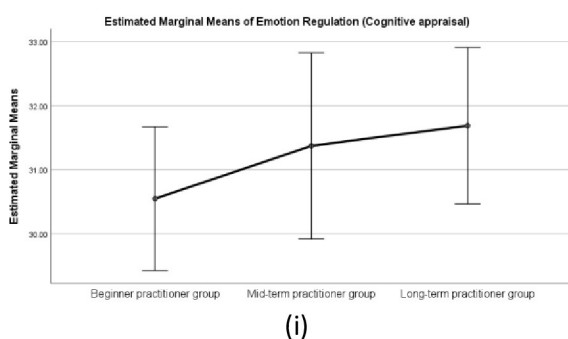
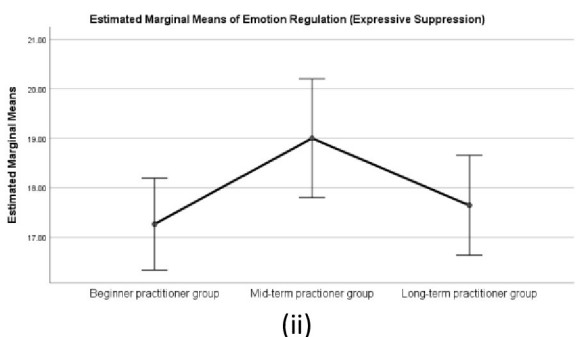

**Fig 6.** Means Plot for Emotion regulation i) Cognitive appraisal ii) Expressive suppression for three practitioner groups. Note: Error bars at 95% Cl.

emotion regulation through cognitive reappraisal strategies. There was no statistically significant difference between the mid-term practitioner group and beginner group p>0.05. Means plot is shown in Fig 6.

## Discussion and conclusions

The aim of present research was to study the effect of practice of yoga and other spiritual practices on the illness perception, wellbeing measures and emotion regulation strategies for adults during COVID19 lockdown. Additionally, the effect of demographic variables such as age, gender, qualification, working status and place of residence was also analyzed. The confirmatory factor analysis confirmed the factor structure for the study sample, which strengthens the findings of this research.

The results examining the demographic variables demonstrate that females reported higher illness concern and were emotionally more impacted than males by the COVID19 lockdown. This finding is in line with an earlier review of literature that reported females to be twice more vulnerable to stress than males in conditions or events of adversity [16, 17, 19]. A recent study investigating depression, anxiety and stress due to COVID19 also found higher stress among females than males [42]. The reason for higher stress in females can be argued to be partially due to increase in the household chores in the absence of any house helps during lockdown, specially for those also managing their professional work. Further, it was also found that younger participants from age group 18–25 years reported feeling more depressed, and had lower peace of mind than older participants. A stressful situation such as fear of losing a job, uncertainty about the future can trigger anxiety, depression and is believed to affect the peace of mind. The rationale for the association of stress and age is given by a study investigating differences in coping strategies across lifespan. The study suggests that older adults use coping strategies that are indicative of greater impulse control and they tend to evaluate conflict situations more positively than younger adults [16–18]. In another study it was found that older adults had lower levels of psychological distress and better dispositional coping compared to younger adults [43]. Perhaps the fear arising from uncertainties was dealt more efficiently by the older population, such that it affected their wellbeing positively. Interestingly, in this study the urban population reported higher perception of the COVID19 consequences than reported by the participants from rural or semi-urban areas. Perhaps, the urban population felt that COVID19 lockdown is going to affect them more adversely than the rural or semi-urban. One of the reasons for this difference could be the job insecurity. Another plausible explanation for the difference in perception of the COVID 19 consequence can be derived from research that states that urban populations are reported more prone to psychological distress than their

rural counterparts [44]. Since urban population consists of the majority of service class which is dependent on their jobs for livelihood, they are more likely to perceive graver consequences of COVID19 lockdown than their rural counterparts which comprise mostly of self- employed people.

The results examining the effect of yoga practice demonstrate that yoga practitioners perceived having higher personal control, higher coherence/understanding, lower emotional impact, lower risk and higher preventive control for contracting COVID19 than other spiritual practitioners and non-practitioners. A number of studies have reported physical and mental health benefits of yoga practice [45–47]. A healthy individual is found to perceive lower cognitive and emotional effects of the illness and a higher preventive control over the illness [9]. On the other hand, the participants who negatively perceived the COVID19 effects experienced greater levels of stress, anxiety or depression and lower wellbeing, also reported in a study on cancer patients [48]. Additionally, in light of the findings of the previous study, the notion that an individual is following a treatment or preventative control therapy positively affects the perception about how well the illness is understood and a sense of personal control over the illness [10]. In this study also yoga practitioners reported to have a better understanding and higher personal control over COVID19. Perhaps yoga practitioners felt that yoga is an effective therapy to cope with COVID19 both for physical as well as mental health.

In this study, it was also found that yoga practitioners had lower depression, lower stress, lower anxiety, higher wellbeing, and higher peace of mind than the other spiritual practitioners and non-practitioner group. Interestingly, the other spiritual practitioners were also found to have a significantly higher peace of mind than the non-practitioners. The other spiritual practitioner group also reported lower depression, anxiety, stress and higher wellbeing than the non-practitioner group, however the difference was not found to be statistically significant in this study. Possibly the other spiritual practices; reading Holy Scriptures and rendering *seva* (selfless service) to the needy and destitute provided solace and peace of mind. A previous study has also reported a positive association between reading scriptures and positive affect and *sukha* (happiness) and a negative association with negative affect and *dukkha* (unhappiness) [32]. As for the non-practitioner group, participants that reportedly followed none of the yoga or spiritual practices, also reported the highest mean score of depression, anxiety and stress and lowest wellbeing and peace of mind.

Results showed that there was no significant difference in resilience among the yoga practitioners, other spiritual practitioners and non-practitioner group. Resilience has for long been debated by researchers to be a trait construct. In this study also resilience was found to be perhaps a more trait-like construct that unfolds over time in response to internal strengths and external supports across lifespan that foster positive outcomes in the face of adversity.

In this study, a significant effect of duration of practice was found on illness perception, and wellbeing related measures. Long term practitioners reported higher personal control and lower illness concern in contracting COVID19 than the mid-term or beginner group. The long- term and mid-term practitioners also reported perceiving lower emotional impact of COVID19 and lower risk in contracting COVID19 than the beginners. The general wellbeing was reported higher by the long term and mid- term practitioners than the beginners group. Further, the long term practitioners were found to have highest peace of mind, lowest depression and anxiety with no significant difference in the mid-term and the beginner group. In an earlier study, sustained practice of yoga is reported to enhance physical strength, promote and improve respiratory and cardiovascular function. The improved physiological functions are believed to reduce stress, anxiety, depression, and enhance overall well-being. In line with the outcomes from this study, the regular practice has also been argued to lead changes in life perspective, self-awareness, a sense of balance between body and mind and generally a positive

outlook to life that maintains general wellbeing even in difficulties [49, 50]. Interestingly, in this study the beginner group, which had started practicing yoga during COVID19 lockdown, reported comparable mean scores of wellbeing and peace of mind with the mid-term practitioner groups. When compared with the non-practitioner group, the beginner group also had lower depression, anxiety, stress and higher wellbeing, peace of mind. Perhaps the routine practice of yoga helped the beginner practitioners to calm the mind and maintain a positive disposition during difficult times of COVID19 lockdown. The outcomes reveal that yoga practice helps in illness perception about COVID19 such that the long- term practitioners feel a better sense of preventive control with a notion of being less prone to contracting COVID19. This perception of lesser vulnerability and a better sense of control over COVID19 is argued to generate lesser stress problems and promote higher wellbeing. The emotion regulation strategy of cognitive reappraisal is further argued to breed a balanced and coherent understanding about the COVID19. The balanced representation of the unknown is argued to tone down the fear factor due to uncertainties caused by COVID19 lockdown thus decreasing the stress, anxiety and depression. Such a state of mind allows one to view an adverse situation with a more pragmatic approach and helps in maintaining a peaceful disposition.

Altogether, the findings from this study shows that yoga is found to be an effective way to manage the stress, anxiety and depression due to and during COVID19 lockdown. The evidence further supports that yoga could be used as a complementary and alternative therapy for the stress related problems due to COVID19. It may also help health practitioners in further promoting yoga-based interventions to facilitate the self-management of the mental health issues due to COVID19.

## Supporting information

**S1 Table. Correlations.**
(DOCX)

## Acknowledgments

The authors appreciates all those who participated in the study and helped to facilitate the research process.

## Author Contributions

**Conceptualization:** Pooja Swami Sahni, Kamlesh Singh, Nitesh Sharma, Rahul Garg.

**Data curation:** Pooja Swami Sahni, Nitesh Sharma.

**Formal analysis:** Pooja Swami Sahni, Kamlesh Singh, Nitesh Sharma.

**Investigation:** Pooja Swami Sahni.

**Methodology:** Pooja Swami Sahni.

**Project administration:** Pooja Swami Sahni, Kamlesh Singh.

**Resources:** Kamlesh Singh.

**Software:** Pooja Swami Sahni.

**Supervision:** Kamlesh Singh, Rahul Garg.

**Validation:** Pooja Swami Sahni.

**Writing – original draft:** Pooja Swami Sahni.

**Writing – review & editing:** Pooja Swami Sahni, Kamlesh Singh, Nitesh Sharma, Rahul Garg.

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
