## [Decision Letter · Decision Letter 0]

2 Nov 2020

PONE-D-20-30379

Yoga an effective strategy for self-management of stress-related problems and wellbeing during COVID19 lockdown: A cross-sectional study

PLOS ONE

Dear Dr. Sahni,

Thank you for submitting your manuscript to PLOS ONE. After careful consideration, we feel that it has merit but does not fully meet PLOS ONE’s publication criteria as it currently stands. Therefore, we invite you to submit a revised version of the manuscript that addresses the points raised during the review process.

We look forward to receiving your revised manuscript.

Kind regards,

Gagan Deep

Academic Editor

PLOS ONE

Journal Requirements:

Reviewers' comments:

Reviewer's Responses to Questions

**Comments to the Author**

1. Is the manuscript technically sound, and do the data support the conclusions?

Reviewer #1: Yes

Reviewer #2: Yes

2. Has the statistical analysis been performed appropriately and rigorously? 

Reviewer #1: Yes

Reviewer #2: Yes

3. Have the authors made all data underlying the findings in their manuscript fully available?

Reviewer #1: Yes

Reviewer #2: Yes

4. Is the manuscript presented in an intelligible fashion and written in standard English?

Reviewer #1: Yes

Reviewer #2: Yes

5. Review Comments to the Author

Reviewer #1: Comments to Authors:

Ref: PONE-D-20-30379

The article entitled ‘Yoga an effective strategy for self-management of stress-related problems and wellbeing during COVID19 lockdown: A cross-sectional study’ by Sahni et al., is an interesting study on the impact of yoga practices on the psychological management of youths in a pandemic related broader societal lockdown that minimizes social interaction. By nature, humans are social animals and have evolved various social institutions to satisfy their psychological well being and thereby overall mental health of its individual members. The lockdown like condition goes against this concept and isolate the individuals, albeit in many cases with their families, in solitary habitation leading to severe mental stress and venting avenues. Thus, in this background, the present manuscript dwells on a pertinent subject.

The introduction is lengthy and can be drastically shortened to remain focused on the problem being studied and the relevant literature in the field. It should be curtailed by half the size (from 4 pages to 2 pages). The 2nd page and half of the third page (line 64 to 94) of the introduction can be removed, and the remaining text may be realigned by moving the line 120 to 125 right after line 63 to maintain the flow.

Abstract says 64.8 % males, while methods says 64.7%. Although the difference is miniscule, the accuracy of data presentation is compromised.

In assessing Brief Illness Perception did authors have options of neutrality in the Questionnaire. E.g. in 179-182, ‘The emotional representation of COVID19 was assessed by 2 items incorporating negative reactions such as fear, anger, and distress (sample item: How much does thinking about this illness affect you emotionally? e.g. does it make you angry, scared, upset or depressed?).’ did the participants have option ‘it did not affect them emotionally?’. Because many participants may not have been affected or felt the given emotion, but would have to choose one of the provided stress response!

At places the references are cited as number in big parentheses [2], at other cited as authors names followed by Year in small parentheses (names et al., 2020). It should be corrected. At yet other the cited authors have not been given numbers e.g. Gross and John (2003).

Once multivariate analysis of variance (MANOVA) has been abbreviated, MANOVA should be subsequently used.

In the results section data presentation is apt. However, if some of the data could be presented in the chart form that would make it more comprehensible for the readers.

In the discussion section, analysis was lacking in some part. E.g. sentences like ‘..demonstrate that females reported higher illness concern and were emotionally more impacted than males by the COVID19 lockdown’ could be substantiated with analysis of the likely causes of more emotional impact. Was the trend same for rural vs urban, young vs old, working vs housewife? And how much did Yoga contributed to cope up with this?

Can the authors comment on whether the forced practice vs voluntary practice having positive effects especially in youths and did the author have data on how many of the practitioners started the yoga voluntarily and how many forced to do by others in the family.

Was the participant being in a family a factor in overall wellbeing vs. a person residing alone during the lockdown?

The manuscript is written well but at some places the excessive use of ‘the’ and other minor errors in sentence formation can be edited carefully.

Reviewer #2: In the present study Sahni et al have analyzed the results of cross-sectional study of yoga practice as an effective strategy for self-management of stress-related problems and well-being. Many statistical methods have been used to present the results pertaining to illness perception, and wellbeing of healthy adults across age, gender, place of work and residence.

The study and study timeline of 4-10 weeks of lockdown due to COVID19 outbreak is appropriate and timely. However, explanation is required in the manuscript to support the statistical analysis especially with respect to:

1.) Daily practice of Yoga with respect to frequency and duration specifically how many days a week and how long (30 minutes, an hours) for all three groups

2.) Higher stress among female during COVID19 lockdown as compared to men

6. PLOS authors have the option to publish the peer review history of their article (what does this mean?). If published, this will include your full peer review and any attached files.

Reviewer #1: No

Reviewer #2: **Yes: **Sangeeta Singh

---

## [Author Response · Author response to Decision Letter 0]

1 Dec 2020

The authors thank the editors and reviewers for recognizing the relevance of the study in the current pandemic times and for their encouraging words. 

The authors also appreciate the efforts of the reviewers and editors for thorough review of the manuscript which is evident from the valuable and constructive feedback that is provided for enriching the manuscript.

Please find below the pointwise response to the reviewers comments:

The article entitled ‘Yoga an effective strategy for self-management of stress-related problems and wellbeing during COVID19 lockdown: A cross-sectional study’ by Sahni et al., is an interesting study on the impact of yoga practices on the psychological management of youths in a pandemic related broader societal lockdown that minimizes social interaction. The authors thank the editors and reviewers for recognizing the relevance of the study in the current pandemic times and for their encouraging words. 

response: The authors also appreciate the efforts of the reviewers and editors for thorough review of the manuscript which is evident from the valuable and constructive feedback that is provided for enriching the manuscript.

 By nature, humans are social animals and have evolved various social institutions to satisfy their psychological well being and thereby overall mental health of its individual members. The lockdown like condition goes against this concept and isolate the individuals, albeit in many cases with their families, in solitary habitation leading to severe mental stress and venting avenues. Thus, in this background, the present manuscript dwells on a pertinent subject. 

The authors completely agrees that the disruptions of the social interactions may be argued to have a detrimental effect on psychological wellbeing. Further the change in social situation posed by pandemic may have led to severe mental stress in many individuals.

1) The introduction is lengthy and can be drastically shortened to remain focused on the problem being studied and the relevant literature in the field. It should be curtailed by half the size (from 4 pages to 2 pages). The introduction has been reduced after incorporating the changes as suggested in point no. 2.

2) The 2nd page and half of the third page (line 64 to 94) of the introduction can be removed, and the remaining text may be realigned by moving the line 120 to 125 right after line 63 to maintain the flow. Thanks for the valuable suggestion. The changes as recommended have been made to the revised manuscript.

3) Abstract says 64.8 % males, while methods says 64.7%. Although the difference is miniscule, the accuracy of data presentation is compromised. 

The correction has been made in the revised manuscript.

4) In assessing Brief Illness Perception did authors have options of neutrality in the Questionnaire. E.g. in 179-182, ‘The emotional representation of COVID19 was assessed by 2 items incorporating negative reactions such as fear, anger, and distress (sample item: How much does thinking about this illness affect you emotionally? e.g. does it make you angry, scared, upset or depressed?).’ did the participants have option ‘it did not affect them emotionally?’. Because many participants may not have been affected or felt the given emotion, but would have to choose one of the provided stress response! The participants were asked to respond to the question,’ How much does thinking about this illness affect you emotionally? e.g. does it make you angry, scared, upset or depressed?’ on the scale presented as linear scale as, ‘ 0 (not at all affected emotionally) to 10 (extremely affected emotionally)’. The participants had the option to choose ‘0’ which meant that ‘it did not affect them emotionally’.

5) At places the references are cited as number in big parentheses [2], at other cited as authors names followed by Year in small parentheses (names et al., 2020). It should be corrected. The citations have been made as numbers in the big parentheses in the revised manuscript as per the PLOS ONE guidelines for referencing.

6) At yet other the cited authors have not been given numbers e.g. Gross and John (2003).

 The corrections in the citations have been made in the revised manuscript.

7) Once multivariate analysis of variance (MANOVA) has been abbreviated, MANOVA should be subsequently used. Multivariate analysis of variance has been abbreviated as MANOVA in all subsequent use in the revised manuscript.

8) In the results section data presentation is apt. However, if some of the data could be presented in the chart form that would make it more comprehensible for the readers.

 The authors had a deep discussion on this proposed form of data representation. The figures 1-6 and table 3 and table 5 provides a quick glimpse of the key outcomes of the study. However, it was felt that many layers of the analysis as described in the manuscript would be compromised if the data representation were to be further reduced. 

9) In the discussion section, analysis was lacking in some part. E.g. sentences like ‘..demonstrate that females reported higher illness concern and were emotionally more impacted than males by the COVID19 lockdown’ could be substantiated with analysis of the likely causes of more emotional impact. In the present study the likely causes of stress were analyzed through an additional item on the three most important causal factors (in rank order) of stress during lockdown (‘Please list in rank-order the THREE most important factors that you believe are reasons for stress due to lockdown’). The responses were analyzed through a percentage analysis of the most frequently reported reasons for stress. 

Additionally, in the revised manuscript (line 387-390) we have also included the gender wise analysis of the reported reasons for stress. 

The authors also feel that a gender specific study could be undertaken in future to further substantiate the causes of emotional impact due to COVID 19 lockdown.

10) Was the trend same for rural vs urban, young vs old, working vs housewife? And how much did Yoga contributed to cope up with this? Yes, the trend for place of residence, age and working status was analyzed, as described in section; Results>demographic Variables. MANOVA results indicated no statistically significant main effect of working status and qualification on the COVID19 perception, wellbeing measures (DASS, SGWB, POMS, and BRS) or emotion regulation strategies (ERQ). There was a significant effect of place of residence on illness consequence of COVID19 F (1,634) =5.61, p<0.05, partial eta squared=-.01, with urban participants reporting higher concern for consequences of COVID19 (M=7.76, SD=5.34) than semi-urban (M=6.95, SD=2.55) or rural (M=7.22, SD=2.88). 

Age had a significant effect on depression (DASS-D) F (1,634) = 9.34, p<0.005, partial eta squared=-.01 and Peace of Mind (POMS) F (1,634) =13.02, p<0.001, partial eta squared=-.02, with participants from age group 18-25 years reporting higher depression (DASS-D)(M=0.97, SD=0.70) than age group 26-35 years (M=0.81, SD=.60), age group 36-45 years (M=0.64, SD=0.62) and age group 46 and above ( M=0.61, SD=0.57).

11) Can the authors comment on whether the forced practice vs voluntary practice having positive effects especially in youths and did the author have data on how many of the practitioners started the yoga voluntarily and how many forced to do by others in the family. The study of forced practice vs voluntary practice is an interesting research question, especially for youth. However, this study did not collect data on whether the respondents practiced yoga of their own will or were forced as a part or compulsory course or family/peer pressure. This stimulating research question that can be addressed by a future research.

12) Was the participant being in a family a factor in overall wellbeing vs. a person residing alone during the lockdown? The authors agree that stress during lockdown would differ for a person staying alone and in a family setup. The stress due to being alone or in a family setup was not one of the factors taken in to account. The focus of the study was to understand the effects of yoga practice on stress during lockdown. 

The authors also feel that the study of wellbeing due to living alone or in a family is a very relevant and important research that should be taken up in future. 

13) The manuscript is written well but at some places the excessive use of ‘the’ and other minor errors in sentence formation can be edited carefully. The revised manuscript has again been reviewed carefully to omit any minor errors.

 Reviewer #2: 

In the present study Sahni et al have analyzed the results of cross-sectional study of yoga practice as an effective strategy for self-management of stress-related problems and well-being. Many statistical methods have been used to present the results pertaining to illness perception, and wellbeing of healthy adults across age, gender, place of work and residence. The study and study timeline of 4-10 weeks of lockdown due to COVID19 outbreak is appropriate and timely. The authors are thankful to the reviewers to have recognized the in depth statistical analysis conducted for the study.

The authors are also grateful to the editor and reviewers for acknowledging the relevance of the manuscript for the current pandemic situation. 

15) However, explanation is required in the manuscript to support the statistical analysis especially with respect to:

Daily practice of Yoga with respect to frequency and duration specifically how many days a week and how long (30 minutes, an hours) for all three groups In the study sample, 59.6% (n=384) reported that they practice yoga and were categorized as yoga practitioners. Further, among the yoga practitioners, 35% (n=134) were beginners (those who had started yoga practice during COVID19 lockdown period), 39.7% (n=152) were mid- term (1≤year of practice≤4), and 25.32 % (n=97) were Long term (≥ 5 years of practice) practitioners. 

“Within the beginners, 39.9 % reported practicing yoga for all days in the week, 23.9 % for 5-6 days, 23.2 % for 3-4 days and 13 % for 1-2 days in a week. For mid-term practitioners 32.4% reported practicing yoga for all days in the week, 30.4 % for 5-6 days, 29.7 % for 3-4 days and 7.4 % for 1-2 days in a week. For long term practitioners, 58.8 % reported practicing yoga for all days in the week, 19.6 % for 5-6 days, 11.3 % for 3-4 days and 10.3 % for 1-2 days in a week.

The data for frequency in terms of hours of practice per day was asked as an open choice question,’ How many minutes in a day do you practice yoga? Beginners reported on an average spending 31.2 mins, mid-term practitioner average of 39.10mins, and long term practitioners reported spending average of 51.2 mins, on their daily yoga practice.”

This above quoted text has been added to the revised manuscript.

16) Higher stress among female during COVID19 lockdown as compared to men The explanation for the higher stress among female during COVID19 lockdown as compared to men has been added in the revised manuscript as described in point no. 9 of this document.

Additional remarks have also been added to discussion section.

---

## [Decision Letter · Decision Letter 1]

26 Dec 2020

Yoga an effective strategy for self-management of stress-related problems and wellbeing during COVID19 lockdown: A cross-sectional study

PONE-D-20-30379R1

Dear Dr. Sahni,

We’re pleased to inform you that your manuscript has been judged scientifically suitable for publication and will be formally accepted for publication once it meets all outstanding technical requirements.

Kind regards,

Gagan Deep

Academic Editor

PLOS ONE

Additional Editor Comments (optional):

Reviewers' comments:

Reviewer's Responses to Questions

**Comments to the Author**

1. If the authors have adequately addressed your comments raised in a previous round of review and you feel that this manuscript is now acceptable for publication, you may indicate that here to bypass the “Comments to the Author” section, enter your conflict of interest statement in the “Confidential to Editor” section, and submit your "Accept" recommendation.

Reviewer #1: All comments have been addressed

Reviewer #2: All comments have been addressed

2. Is the manuscript technically sound, and do the data support the conclusions?

Reviewer #1: Yes

Reviewer #2: Yes

3. Has the statistical analysis been performed appropriately and rigorously? 

Reviewer #1: Yes

Reviewer #2: Yes

4. Have the authors made all data underlying the findings in their manuscript fully available?

Reviewer #1: Yes

Reviewer #2: Yes

5. Is the manuscript presented in an intelligible fashion and written in standard English?

Reviewer #1: Yes

Reviewer #2: (No Response)

6. Review Comments to the Author

Reviewer #1: The authors have responded satisfactorily to all the comments by this reviewer and have revised the manuscript accordingly. The manuscript may be accepted for publication.

Reviewer #2: (No Response)

7. PLOS authors have the option to publish the peer review history of their article (what does this mean?). If published, this will include your full peer review and any attached files.

Reviewer #1: **Yes: **Umesh C S Yadav

Reviewer #2: No

---

## [Editor Report · Acceptance letter]

19 Jan 2021

PONE-D-20-30379R1 

Yoga an effective strategy for self-management of stress-related problems and wellbeing during COVID19 lockdown: A cross-sectional study 

Dear Dr. Sahni:

I'm pleased to inform you that your manuscript has been deemed suitable for publication in PLOS ONE. Congratulations! Your manuscript is now with our production department. 

Kind regards, 

on behalf of

Dr. Gagan Deep 

Academic Editor

PLOS ONE